# β-Catenin-Specific Inhibitor, iCRT14, Promotes BoHV-1 Infection-Induced DNA Damage in Human A549 Lung Adenocarcinoma Cells by Enhancing Viral Protein Expression

**DOI:** 10.3390/ijms23042328

**Published:** 2022-02-19

**Authors:** Xiuyan Ding, Weifeng Yuan, Hao Yang, Chang Liu, Shitao Li, Liqian Zhu

**Affiliations:** 1College of Life Science, Institute of Life Science and Green Development, Hebei University, Baoding 071002, China; yding202201@163.com (X.D.); yanghao11yyy@163.com (H.Y.); liuc990728@163.com (C.L.); 2Jiangsu Co-Innovation Center for Prevention and Control of Important Animal Infectious Diseases and Zoonoses, College of Veterinary Medicine, Yangzhou University, Yangzhou 225009, China; 3Institute of Animal Sciences, Chinese Academy of Agricultural Sciences, Beijing 100193, China; yuanweifeng@caas.cn; 4Department of Microbiology and Immunology, Tulane University, New Orleans, LA 70118, USA; sli38@tulane.edu; 5Key Laboratory of Microbial Diversity Research and Application of Hebei Province, College of Life Science, Hebei University, Baoding 071002, China

**Keywords:** BoHV-1, β-catenin, 53BP1, γH2AX, DNA damage

## Abstract

Oncolytic bovine herpesvirus type 1 (BoHV-1) infection induces DNA damage in human lung adenocarcinoma cell line A549. However, the underlying mechanisms are not fully understood. We found that BoHV-1 infection decreased the steady-state protein levels of p53-binding protein 1 (53BP1), which plays a central role in dictating DNA damage repair and maintaining genomic stability. Furthermore, BoHV-1 impaired the formation of 53BP1 foci, suggesting that BoHV-1 inhibits 53BP1-mediated DNA damage repair. Interestingly, BoHV-1 infection redistributed intracellular β-catenin, and iCRT14 (5-[[2,5-Dimethyl-1-(3-pyridinyl)-1H-pyrrol-3-yl]methylene]-3-phenyl-2,4-thiazolidinedione), a β-catenin-specific inhibitor, enhanced certain viral protein expression, such as the envelope glycoproteins gC and gD, and enhanced virus infection-induced DNA damage. Therefore, for the first time, we provide evidence showing that BoHV-1 infection disrupts 53BP1-mediated DNA damage repair and suggest β-catenin as a potential host factor restricting both virus replication and DNA damage in A549 cells.

## 1. Introduction

Bovine herpesvirus type 1 (BoHV-1) and herpes simplex virus-1(HSV-1) are members of the family Herpesviridae and the subfamily Alphaherpesvirinae [1]. BoHV-1 infects cattle and induces lesions on mucosal surfaces, genital tracts, and nervous systems [2]. Although BoHV-1 cannot infect humans, it is a novel oncolytic virus [3] that can infect numerous human tumor cells from different origins, indicating a broad spectrum of oncolytic effects [4]. Given the safety to human beings, along with the broad oncolytic spectrum, BoHV-1 is an attractive candidate for the development of virotherapy targeting diverse cancers.

DNA damage poses a constant threat to cells. If left unrepaired, it may give rise to deleterious mutations and genome instability, consequently leading to cell death or disease [5]. Double-strand breaks (DSBs) are considered to be among the most hazardous forms of DNA damage, severely compromising genome stability [6]. Phosphorylation of the variant histone H2AX at Serine139 (γH2AX) is an early cellular response to the induction of DSBs. Detection of γH2AX is often considered a sensitive molecular marker for monitoring DNA damage and repair [7]. In addition, an alkaline version of a comet assay is also a sensitive method to assess DSBs in individual cells [8]. With comet assays, we have previously revealed that BoHV-1 has infected lung adenocarcinoma A549 cells and induced oxidative DNA damage in human lung carcinoma A549 cells [9,10]. However, the mechanism of how the DNA damage was induced in the virus-infected tumor cells remains to be characterized.

Accumulating studies have indicated that tumor suppressor p53-binding protein 1 (53BP1) plays an essential role in the maintenance of genome integrity and stability by orchestrating the repair of double-strand breaks (DSBs) [11]. For DSB repair, 53BP1 must be recruited on DSBs’ flanking chromatin, where 53BP1 foci are formed together with other critical molecules to initiate DNA repair via nonhomologous end-joining (NHEJ) pathways [12]. In response to DSBs, phosphorylation of 53BP1 and ubiquitylation of histone H2A is stimulated via complicated mechanisms triggered by the ataxia-telangiectasia mutated (ATM) protein kinase, which is necessary for both 53BP1 recruitment and retention around DSBs [13]. The retention of 53BP1 at DSBs requires γH2AX, which colocalizes with 53BP1 and forms foci [14]. The interplay between 53BP1 and BoHV-1 infection in tumor cells has not been reported.

β-catenin is involved in the regulation of multiple cellular functions, such as adhesion and gene transcription. Mutations and overexpression of β-catenin are closely associated with many cancers, including lung cancers [15]. Thus, β-catenin has been implicated for therapeutic interventions in various cancers. It has been reported that β-catenin was potentially involved in the regulation of BoHV-1 productive infection in bovine kidney (CRIB) cells and latency in trigeminal ganglia [16,17]. In addition, it was reported that there was crosstalk between DNA damage response and β-catenin [18]. For example, it has been reported that β-catenin signaling promoted DNA damage repair by transactivation of the high-mobility group box 1 protein (HMGB1) in esophageal squamous cell carcinoma [19]. β-catenin may mediate the chemoradiotherapy resistance of colorectal cancer cells by promoting DNA damage repair [20]. Thus, β-catenin has been implicated in DNA damage repair in numerous tumor cells. It has been reported that β-catenin-dependent c-myc expression by HCV nonstructural protein NS5A enhanced DNA damage, which was potentially involved in HCV-associated hepatocellular carcinoma. It seems that the HCV NS5A protein creates β-catenin-compromising DNA damage. However, whether β-catenin is involved in the virus productive infection and virus infection-induced DNA damage in A549 cells remains to be resolved.

In this report, we demonstrated that BoHV-1 infection decreased 53BP1 expression and reduced 53BP1 foci formation. Moreover, for the first time, we found that β-catenin is involved in the restriction of viral protein expression and virus-induced DNA damage in A549 cells.

## 2. Results

### 2.1. BoHV-1 Infection Induced γH2AX foci, a Hallmark of DNA Damage in A549 Cells 

In response to DSBs induced by many toxic elements, such as X-rays, γ-radiation, and UV-light irradiation, γH2AX levels increase and accumulate in the DSBs to develop specific foci under fluorescence microscopy, which are generally considered molecular markers of DNA damage [7]. Notably, whether these indicators are also suited for the analysis of BoHV-1 infection-induced DNA damage in A549 cells has not been validated. To test whether γH2AX could serve as a consistent marker of DNA damage in the context of BoHV-1 infection, we initially examined γH2AX protein levels following virus infection of A549 cells. At 36 h after infection, higher γH2AX protein levels were detected in virus-infected A549 cells, which increased approximately two-fold relative to the control, while obvious change was not observed at 24 h post-infection (hpi) (Figure 1A). With a comet assay, we found that the comet tails, indicators of DSBs, were readily observed after infection at both 24 and 36 hpi (Figure 1B). The extent of the DNA damage was evaluated by calculating the ratio of DNA fluorescence in the tail to that in the whole cell (tailDNA%). The tailDNA% was 6.50% in mock-infected cells, which increased to 12.64% and 24.11% after infection for 24 and 36 h, respectively (Figure 1C). 

The expression of viral glycoprotein gC was detected at 24 and 36 hpi (Figure 1D). In support of these observations, evident effects of virus-induced cytopathology were observed at 24 and 36 hpi when the cell morphology was examined under microscopy (Figure 1E). These data confirmed that productive viral infection occurred at 24 hpi, even though increased γH2AX protein levels were not observed. Given that DSBs could be detected by comet assay at 24 hpi and that comet assays are a sensitive and reliable method to measure DNA damage from DSBs, we suggest that γH2AX protein levels detected by Western blot analysis are not a robust indicator of DNA damage induced by BoHV-1 infection before 24 hpi.

Counting the number of γH2AX foci by fluorescence microscopy is a useful method to assess the extent of DSBs quantitatively [21]. Thus, an immunofluorescence (IFA) assay was performed to test the formation of γH2AX foci after BoHV-1 infection for 24 h. As a result, nuclear γH2AX foci were detected in a subset of mock-infected A549 cells, and more foci were observed at a higher frequency following virus infection (Figure 2A). Apart from the nucleus, a subset of γH2AX foci was found to be located at the cytosol in certain cells following infection, indicating that virus infection leads to the re-localization of γH2AX (Figure 2B). Given that nuclear γH2AX foci are essentially associated with DNA damage repair, only the nuclear foci were collected for subsequent analysis. Quantitative analysis indicated that the steady-state percentage of cells with ≥10 foci within an individual cell was 8.21% in mock-infected cells, which significantly increased to 23.86% following virus infection (Figure 2C). Therefore, the percentage of cells harboring ≥10 foci was increased approximately three-fold by virus infection, indicating that DNA damage was induced, which was consistent with the comet assay data demonstrated in Figure 1A. 

Taking these data together, the assessment of γH2AX foci under fluorescent microscopy can be used for the examination of DNA damage induced by BoHV-1 infection in A549 cells.

### 2.2. BoHV-1 Infection Leads to Inhibition of 53BP1 Signaling

53BP1-mediated NHEJ events play a critical role in the repair of DSBs [22]. Recruitment of 53BP1 into DSBs to form foci is essential for repair. To understand whether 53BP1-mediated DNA damage repair was influenced, we initially examined the intracellular distribution of 53BP1 following infection of the cultured cells. An IFA assay indicated that, following virus infection, 53BP1 foci with bigger size and highlighted staining generally observed in uninfected cells were rarely detected following virus infection (Figure 3A). After infection for 36 h, the percentage of cells with 53BP1 foci reduced to 18.9%, while still nearly 41% in the uninfected controls (Figure 3B), indicating the impaired ability of 53BP1 to form foci.

We then detected the steady-state protein expression of 53BP1 by Western blot analysis. At 24 and 36 hpi, 53BP1 protein levels were consistently reduced. Similar to mock-infected cells, 53BP1 protein levels were approximately reduced to 34.1% and 52.6% at 24 and 36 hpi, respectively (Figure 3C). Since virus infection leads to 53BP1 protein depletion, a longer exposure time was required to observe 53BP1 protein staining in the virus-infected cells than in the uninfected cells under the confocal microscope. Thus, we could not compare 53BP1 protein levels in infected vs uninfected cells using IFA images. Collectively, virus infection led to 53BP1 protein depletion and impaired the capacity to form 53BP1 foci. 

### 2.3. BoHV-1 Infection Alters β-Catenin Subcellular Localization

Accumulating studies have indicated that β-catenin signaling had effects on DNA damage response [18] and that β-catenin stimulated BoHV-1 productive infection in cell cultures [16]. Moreover, β-catenin is a potential therapeutic target for various cancers [23]. To understand whether virus infection influences β-catenin signaling, we initially examined β-catenin protein levels following infection of cell cultures. Similar to uninfected cells, β-catenin levels were not altered by virus infection at either 24 or 36 hpi (Figure 4A). A faint band (denoted by an asterisk) with a molecular weight less than that of β-catenin was observed at 36 hpi (Figure 4A). When the immunoblot was developed with overexposure, the faint band was also observed at 24 hpi (Figure 4A, middle panel). It is unlikely that this band is a viral protein because this band was not readily detected in virus-infected MDBK cells (Figure 4B), and the less viral protein gC was observed in virus-infected A549 cells relative to that in virus-infected MDBK cells (Figure 4C). Thus, this band may represent either isoforms or proteolytic products of β-catenin.

IFA assay indicated that, in the mock-infected cells, intracellular β-catenin was evenly distributed in the cytosol and the nucleus and highly expressed on the plasma membrane (Figure 5). However, after BoHV-1 infection, β-catenin was predominantly located in the cytosol and concentrated into spots that had irregular shapes with highlighted staining (Figure 5). Taken together, BoHV-1 infection relocalized intracellular β-catenin and induced β-catenin to form cytosolic speckles.

### 2.4. β-Catenin-Specific Inhibitor iCRT14 Promotes BoHV-1 Infection-Induced DNA Damage 

Our data indicated that the increased formation of γH2AX foci could be used to assess DNA damage induced by BoHV-1 infection. iCRT14 is a known β-catenin-specific inhibitor that inhibits β-catenin-dependent transcription [16]. To understand whether β-catenin has effects on the virus infection-induced DNA damage, we detected the formation of γH2AX foci in the presence of iCRT14. We found that more foci were observed at a higher frequency following iCRT14 treatment (Figure 6A). Quantitative analysis indicated that the percentage of cells having ≥10 foci within an individual cell was 26.64% in mock-treated controls, which significantly increased to 46.25% in the presence of iCRT14 (Figure 6B). BoHV-1 infection-induced DNA damage was further promoted by the treatment of iCRT14. 

These data suggested that β-catenin has the capacity to restrict DNA damage, and the β-catenin-specific inhibitor iCRT14 promotes DNA damage elicited by virus infection.

### 2.5. β-catenin-Specific Inhibitor iCRT14 Promotes Viral Gene Expression

To understand why iCRT14 promotes DNA damage elicited by virus infection, we investigated whether it had effects on viral gene expression. For this aim, A549 cells infected with BoHV-1 were treated with either DMSO control or iCRT14, and the viral proteins gC and gD were detected. At 24 and 36 h after infection, a gC-specific band was detected in virus-infected cells, which was enhanced when treated with iCRT14 (Figure 7A,B). Quantitative analysis indicated that, similar to the mock-treated control, gC and gD protein levels were increased approximately 2.6- and 2.0-fold by iCRT14, respectively, at 24 hpi (Figure 7A). In agreement with what was observed at 24 hpi, gC and gD protein levels were increased approximately 4.1- and 3.6-fold by iCRT14 at 36 hpi, respectively (Figure 7B). These data suggested that iCRT14 promoted certain viral protein expressions, which might be due to the promoted DNA damage. It is also possible that the promoted viral protein expression by iCRT14 might consequently exacerbate DNA damage.

## 3. Discussion

Exposure to various genotoxic stresses, such as UV treatment, leads to a sharp peak of γH2AX and accumulation of γH2AX at the sites of damaged DNA that forms highlighted foci with IFA assay. γH2AX foci or γH2AX protein levels are widely detected to measure DNA damage [7]. Mechanically, the DNA damage due to BoHV-1 infection is largely different from that induced by exposure to canonical genotoxic reagents (such as UV exposure) because viral products, such as viral proteins and viral nucleic acid, are actively introduced into the nucleus. Unlike UV irradiation, the viral products cannot be retracted from virus-infected cells. By comparison with the data of comet assays, we found that the measurement of γH2AX foci is acceptable to monitor virus infection-induced DNA damage.

Interestingly, we found that a subset of γH2AX foci was also located at the cytoplasm (Figure 2B). Though we could not explain the mechanism underlying the formation of γH2AX foci in the cytoplasm, we speculate that the foci may be associated with mitochondria DNA damage because BoHV-1 infection also induces mitochondria dysfunction in bovine kidney (MDBK) cells [24], which deserves further study in the future.

We found that virus infection led to the depletion of 53BP1 and reduced 53BP1 foci (Figure 3), which may impair 53BP1-mediated DNA damage repair and, consequently, account for virus-induced DNA damage. The depletion of 53BP1 was likely a result of the virus-mediated host shutoff activity because viral proteins, such as bICP27, are able to inhibit host gene expression [25], and bICP0 could promote proteasome-dependent degradation of certain cellular proteins [26]. Given that 53BP1 plays a critical role in DNA damage repair, further study revealing the detailed mechanisms of 53BP1 depletion is an interesting subject that remains to be determined in the future. 

β-catenin stimulates BoHV-1 productive infection in CRIB (BVDV resistant MDBK cells) cells, and the β-catenin-specific inhibitor iCRT14 significantly inhibits productive viral infection in cell cultures [16]. We also found that iCRT14 increased the expression of viral proteins, including gC and gD, suggesting that β-catenin signaling is essentially a host factor restricting virus replication in A549 cells. Redistribution of intracellular β-catenin by virus infection is a potential mechanism to resist β-catenin restrictive activities, although we could not rule out the possibility of a cellular response to restrict virus replication. A549 are human tumor cells. The virus infection in A549 cells demonstrated a large difference in biological characters from MDBK cells. For example, Akt was activated in both A549 and MDBK cells, which is essential for virus productive infection in MDBK cells but not in A549 cells [27]. Furthermore, the protein expression levels of β-catenin were significantly altered in the virus-infected CRIB cells [16] but not in A549 cells (Figure 4A). Similar findings have been reported in Epstein–Barr virus (EBV) infected B-lymphocytes, as β-catenin rapidly degraded in type I B-lymphocytic lines, but it stabilized in type III B cell lines [28]. Taken together, these novel findings further demonstrate that the diverse effects of β-catenin on virus replication may have cell-type-dependent manners.

Moreover, we found that the β-catenin-specific inhibitor iCRT14 enhanced DNA damage (Figure 6), which is consistent with a previous report that inhibition of β-catenin signaling promoted DNA damage induced by benzo[a]pyrene in human colon cancer cells [29]. It has been reported that virus replication products, such as the virus regulatory protein bICP0 and glycoprotein D (gD), are known to have cytotoxicity and have induced cell death [30,31], and the increased expression of viral protein gD by iCRT14 supports the findings that iCRT14 could enhance virus-induced DNA damage.

Of note, β-catenin is overexpressed in diverse tumors, including lung cancers, and it plays an important role in the stimulation of tumorigenesis, invasion, metastasis, and chemotherapy resistance, and, therefore, is regarded as a novel target of cancer therapy. Unexpectedly, our data indicated that BoHV-1 oncolytic activity was not relying on the suppression of β-catenin signaling. In contrast, it seems that β-catenin had limiting effects on viral gene expression. However, our data indicated that a β-catenin-specific inhibitor, such as iCRT14, could efficiently enhance virus infection-induced DNA damage and viral gene expression, raising a possibility that combination therapy with both BoHV-1 and iCRT14 might achieve synergism of antitumor efficacy via the induction of DNA damage, which needs extensive studies in vivo in the future.

In summary, in this study, for the first time, we provided insight into how β-catenin was involved in BoHV-1 productive infection and virus-infection induced DNA damage in human tumor cells. We also provided the first evidence that the β-catenin inhibitor, iCRT14 promoted viral gene expression (gC and gD). Therefore, β-catenin may restrict virus replication and inhibit DNA damage in human lung carcinoma A549 cells. 

## 4. Materials and Methods

### 4.1. Cells and Virus

A549 cells and MDBK cells were both purchased from the Chinese Model Culture Preservation Center (Shanghai, China). They were maintained in DMEM medium supplemented with 10% fetal bovine serum (ThermoFisher Scientific, cat # 10270-106). BoHV-1 (NJ-16-1, isolated in China [32]) was propagated in MDBK cells. Aliquots of virus stock were stored at −70 °C until use.

### 4.2. Antibodies

The antibodies used in this study were as follows: 53BP1 polyclonal antibody (pAb) (Abcam, cat#ab87097), phosporylated-H2A.X (γH2A.X) monoclonal antibody (mAb) (Cell Signaling Technology, cat#2577), β-catenin mAb (Abcam, cat#ab32572), BoHV-1 gC mAb (VMRD, Inc., cat#F2), BoHV-1 gD mAb (VMRD, Inc., cat#1B8-F11), β-Actin mAb (ProteinTech Group, cat#60008-1-Ig), HRP (horseradish peroxidase)-labeled goat anti-mouse IgG (Cell Signaling Technology, cat#7076, 1:3000), HRP-labeled goat anti-rabbit IgG (Cell Signaling Technology, cat#7074), and Alexa Fluor 488^®^-conjugated goat anti-rabbit IgG (H + L) (Invitrogen, cat# A-11008).

### 4.3. Western Blotting Analysis

A549 cells in 60 mm dishes were mock-infected or infected with BoHV-1 at an MOI of 1 for 24, 36, and 48 h. Cells were lysed with an RIPA buffer (1 × PBS, 1% NP-40, 0.5% sodium deoxycholate, 0.1% SDS) that was supplemented with protease inhibitor cocktail [33]. Cell lysates were clarified by centrifugation at 13,000 rpm for 10 min. The clarified cell lysates were collected and boiled in 5X Laemmli sample buffer for 5 min and separated on an 8 or 10% SDS-polyacrylamide gel with approximately 50 μg of protein loaded for each lane. The separated proteins were transferred to a polyvinylidene fluoride (PVDF) membrane. After blocking with 5% skim milk, the membrane was incubated with the primary antibodies described above, followed by HRP-conjugated secondary antibodies in Tris-buffered saline (TBS). After extensive washing with TBS containing 0.05% Tween-20, immune reactive bands were developed after enhanced chemiluminescence (ECL) reaction using a Fusion FX image acquisition system (Vilbur, Marne-la-Vallee cedex 3, France).

The intensity of the detected protein bands was quantitatively analyzed with free ImageJ software. β-Actin was tested, along with individual proteins for protein loading control. The band intensity was initially normalized to β-Actin, and the fold change after infection was calculated. Protein levels in mock-infected cells were set to 1.

### 4.4. Immunofluorescence Assay (IFA)

A549 cells seeded into 2-well chamber slides (Nunc Inc., IL, USA) were mock-infected or infected with BoHV-1 (MOI = 0.1) for 36 h. Cells were fixed with 4% paraformaldehyde prepared in PBS (pH 7.4) for 10 min at room temperature, permeabilized with 0.25% Triton X-100 in PBS (pH 7.4) for 10 min at room temperature, and blocked with 1% BSA in PBST (PBS+ 0.1% Tween-20) for 1 h followed by incubation with the antibodies against γH2A.X (1 : 800 dilution), β-catenin (1 : 800 dilution), and 53BP1 (1 : 500 dilution) in 1% BSA in PBST for 12 h at 4 °C. The cells were washed three times with PBST and incubated with Alexa Fluor 488^®^-conjugated goat anti-rabbit IgG (H + L) (Invitrogen, cat# A-11008, 1: 2000 dilution) for 1 h in the dark. After three washings, nuclei were stained with DAPI (4′,6-diamidino-2-phenylindole). The cells were covered with coverslips using an antifade mounting medium (Electron Microscopy Sciences, Inc., cat# 50-247-04). Images were captured using a confocal microscope (Leica Camera).

### 4.5. Comet Assay

DNA damage was evaluated through an alkaline comet assay (single-cell gel electrophoresis) according to the method described elsewhere with modification [8]. In brief, A549 cells in 24-well plates were infected with BoHV-1 (MOI = 0.1). After virus infection for 24 and 36 h, the cells were collected and suspended in low melting agarose and placed on slides coated with 1% normal melting agarose, and low melting agarose was added as the top layer. Cells were lysed in cold (4 °C) lysis buffer (2.5 M NaCl, 100 mM Na2EDTA, 10 mM Tris, 1% Triton X, and 10% DMSO; pH 10.0) for 1 h. The slides were subjected to horizontal gel electrophoresis in cold (4 °C) alkaline electrophoresis buffer (300 mM NaOH and 1 mM Na2EDTA; pH 12.5) at 25 V and 300 mA for 40 min. The slides were then soaked twice with neutralization buffer (0.4 M Trizma base, pH 7.5, 4 °C) for 10 min and air-dried. DNA was stained with PI (20 μg/mL), and images were subsequently captured using a fluorescence microscope. 

## Figures and Tables

**Figure 1 ijms-23-02328-f001:**
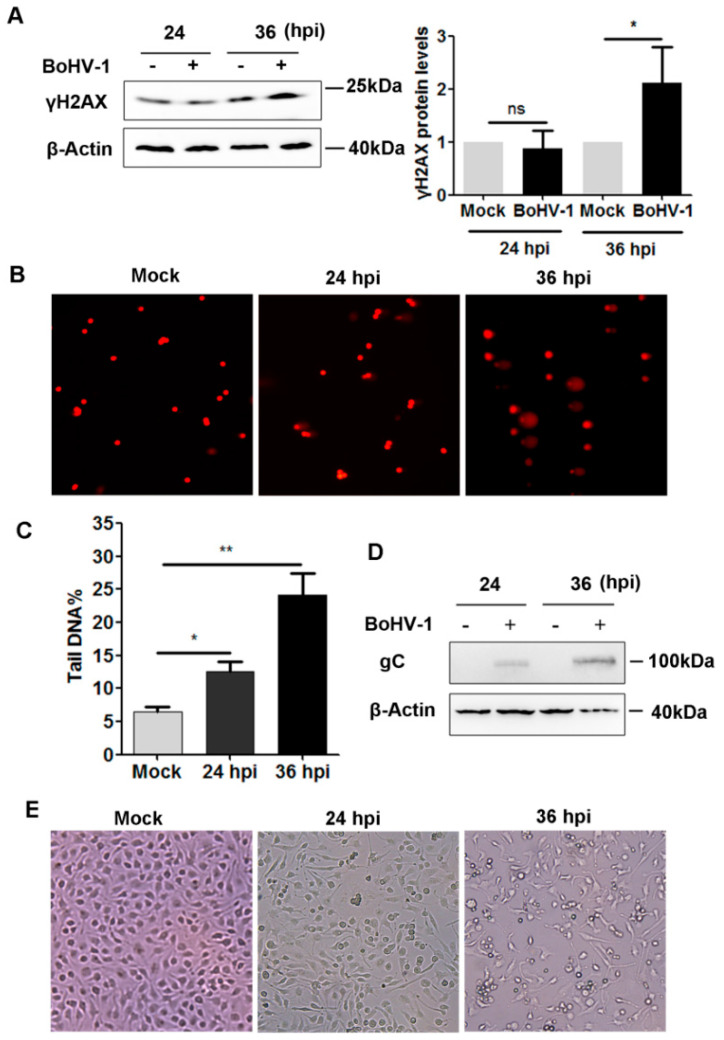
Assessment of γH2AX protein levels as an indicator of BoHV-1 infection-induced DNA damage in A549 cells. (**A**,**D**) Confluent A549 cells in 60 mm dishes were mock-infected or infected with BoHV-1 at an MOI of 0.1 for 24 and 36 h. The cell lysates were prepared and subjected to Western blot analysis to detect the protein levels of γH2AX (**A**) and viral protein gC (**D**). In parallel, β-Actin was tested in the same gel to indicate protein loading. The relative band intensity was analyzed with ImageJ software. The band intensity of γH2AX was initially normalized to β-Actin, and the fold change after infection was calculated by comparison with the uninfected control that was arbitrarily set as 1. Images represent data from three independent experiments. Data shown in the right panel of (**A**) are the means ± SD of the three independent experiments. Statistical analyses were performed using Student’s *t*-test (ns, not significant; * *p* < 0.05). (**B**,**C**,**E**) A549 cells were infected with BoHV-1 (MOI = 0.1) for 24 and 36 hpi. The cells were photographed to compare the effect of virus-induced cytotoxicity to uninfected cells (**E**). Then the cells were collected and subjected to comet assays (**B**). Images represent data of three independent experiments (magnification ×200). Three hundred cells were randomly selected from individual samples for the analysis of TailDNA% with CASP software. Data shown are the means ± SD of the three independent experiments. Statistical analyses were performed using Student’s *t*-test (* *p* < 0.05, ** *p* < 0.01).

**Figure 2 ijms-23-02328-f002:**
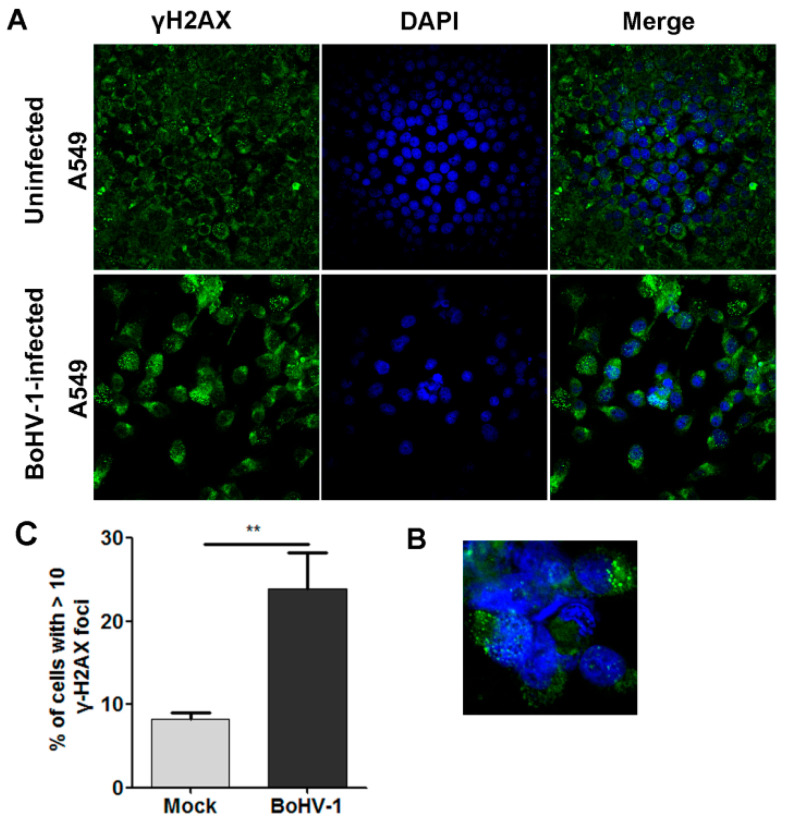
Assessment of γH2AX foci formation as an indicator of BoHV-1-induced DNA damage in A549 cells. (**A**) A549 cells were either mock-infected or infected with BoHV-1 at an MOI of 0.1 for 24 h. After washing three times with PBS, cells were fixed with 4% formaldehyde, and γH2AX was detected by IFA. Nuclei were stained with DAPI. Images were obtained by performing confocal microscopy (magnification ×600 with 2.5 zoom). (**B**) Representative cells with γH2AX localized at the cytoplasm. (**C**) The percentage of cells with ≥10 foci within an individual cell was calculated with over 600 cells counted. The error bars denote the variability between the three independent experiments. Significance was assessed with Student’s *t*-test (** *p* < 0.01).

**Figure 3 ijms-23-02328-f003:**
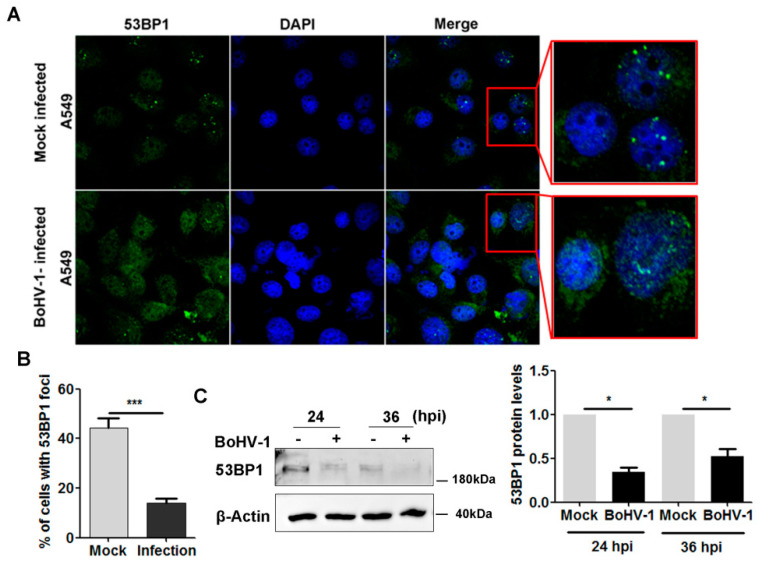
Effects of BoHV-1 infection on 53BP1 expression and foci formation. (**A**) A549 cells were either mock-infected or infected with BoHV-1 at an MOI of 0.1 for 24 h. The cells were fixed with 4% formaldehyde, and 53BP1 was detected by IFA. Nuclei were stained with DAPI. Images were obtained by confocal microscopy (magnification ×600 with 2.5 zoom). (**B**) Approximately 600 cells were counted to calculate the percentage of cells with 53BP1 foci. Error bars denote the variability between the three independent experiments. Significance was assessed with Student’s *t*-test (*** *p* < 0.001). (**C**) After virus infection for 24 h as described in panel A, the cell lysates were prepared for Western blot analysis to detect 53BP1 and β-Actin. The band intensity of 53BP1 was initially normalized to β-Actin, and the fold change after virus infection was calculated by comparison to the uninfected control that was arbitrarily set as 1. Images represent data from three independent experiments. Data shown in the right panel are the means ± SD of the three independent experiments. Statistical analyses were performed using Student’s *t*-test (* *p* < 0.05).

**Figure 4 ijms-23-02328-f004:**
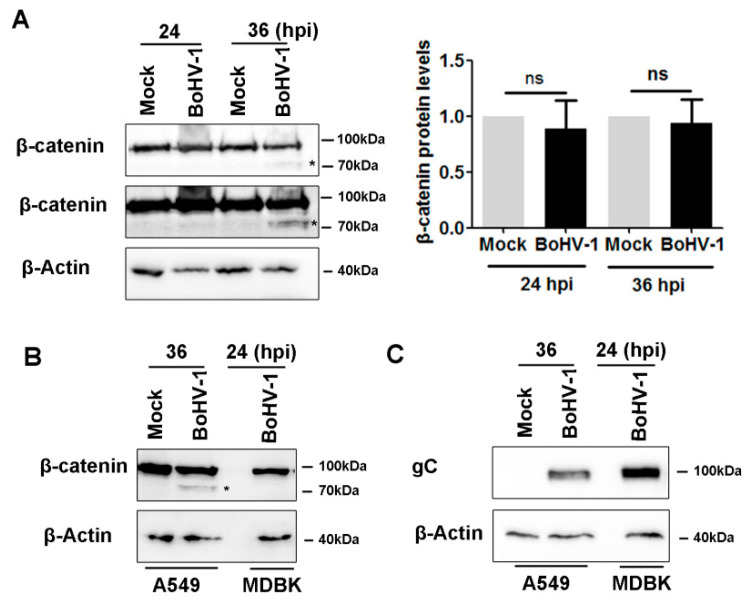
Detection of β-catenin protein levels in response to BoHV-1 infection in A549 cells. (**A**) A549 cells in 60 mm dishes were either mock-infected or infected with BoHV-1 at an MOI of 0.1 for 24 and 36 h. The cell lysates were prepared and subjected to Western blot analysis to detect protein levels of β-catenin. The immunoblots were developed with a normal exposure time (upper panel) and an extra-long exposure time (middle panel). The asterisk (*) denotes a faint band that was detected with the β-catenin antibody. The band intensity of β-catenin was initially normalized to β-Actin, and the fold change after virus infection was calculated by comparison to the uninfected control that was arbitrarily set as 1. Images represent the data from three independent experiments. Data shown in the right panel are the means ± SD of the three independent experiments. Statistical analyses were performed using Student’s *t*-test (ns, not significant; * *p* < 0.05). (**B**,**C**) A549 cells and MDBK cells in 60 mm dishes were either mock-infected or infected with BoHV-1 (MOI = 0.1) for 36 and 24 h. The cell lysates were subjected to Western blot analysis to detect either β-catenin (**B**) or viral protein gC (**C**). The asterisk in panel C denotes a band with a molecular weight less than that of β-catenin that was detected with the β-catenin antibody. Data shown represents the data of three independent experiments.

**Figure 5 ijms-23-02328-f005:**
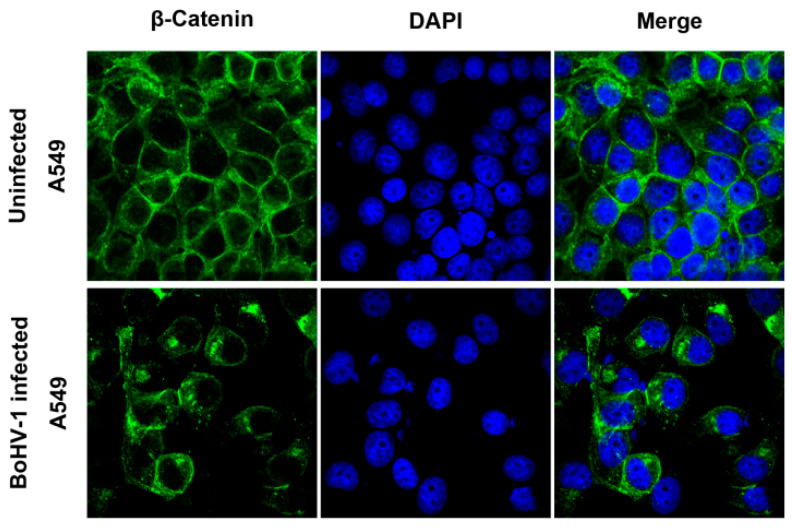
Detection of β-catenin localization in response to BoHV-1 infection in A549 cells. A549 cells were either mock-infected or infected with BoHV-1 at an MOI of 0.1 for 24 h. The cells were fixed with 4% formaldehyde, and β-catenin was detected by IFA. Nuclei were stained with DAPI. Images were obtained by confocal microscopy (magnification ×600 with 2.5 zoom).

**Figure 6 ijms-23-02328-f006:**
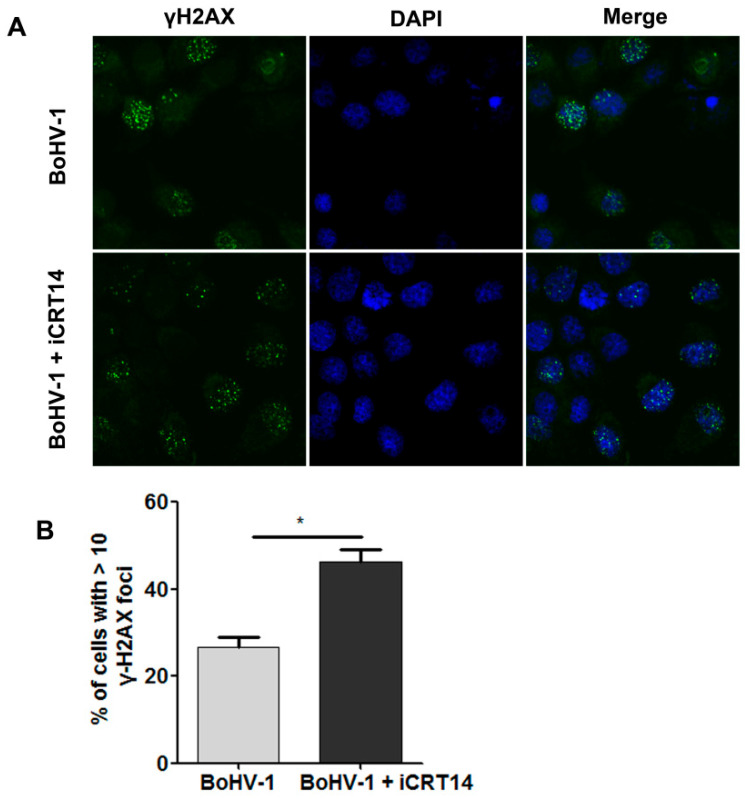
Effects of iCRT14 on BoHV-1-induced formation of γH2AX foci. (**A**) A549 cells infected with BoHV-1 (MOI = 0.1) were either mock-treated with vehicle DMSO or treated with 10 μM of iCRT14. After infection for 24 h, the cells were fixed with 4% formaldehyde, and the formation of γH2AX foci was detected by IFA. Nuclei were stained with DAPI. Images were obtained by confocal microscopy (magnification ×600 with 2.5 zoom). (**B**) The percentage of cells having ≥10 foci within an individual cell was calculated. The error bars denote the variability between the three independent experiments. Significance was assessed with Student’s *t*-test (* *p* < 0.05).

**Figure 7 ijms-23-02328-f007:**
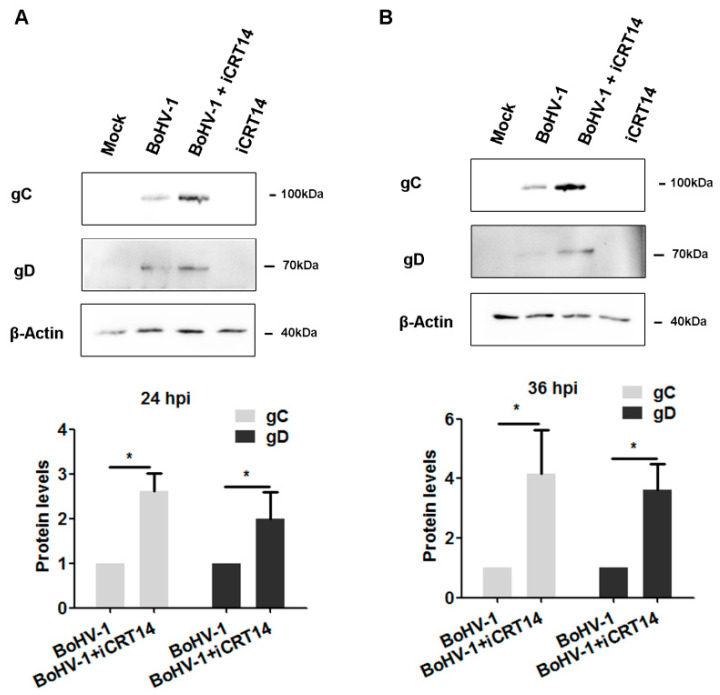
Effects of iCRT14 on BoHV-1 viral protein expression. A549 cells infected with BoHV-1 (MOI = 0.1) were mock-treated with vehicle DMSO or 10 μM of iCRT14. After virus infection for 24 (**A**) and 36 h (**B**), along with treatment by the indicated chemicals, the cell lysates were prepared for Western blot analysis to detect viral proteins gC and gD. The band intensities of both gC and gD were initially normalized to β-Actin, and the fold change after virus infection was calculated by comparison to the uninfected control that was arbitrarily set as 1. Images shown represent the data of three independent experiments. Data shown in the right panel are the means ± SD of the three independent experiments. Statistical analyses were performed using Student’s *t*-test (* *p* < 0.05).

## Data Availability

The authors declare that all the data are available upon request.

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
