# Peer review of "β-Catenin-Specific Inhibitor, iCRT14, Promotes BoHV-1 Infection-Induced DNA Damage in Human A549 Lung Adenocarcinoma Cells by Enhancing Viral Protein Expression"

_ijms, 2022, doi:10.3390/ijms23042328_

Round 1

Reviewer 1 Report

Less than 6 months ago, the laboratory of Liqian Zhu published an interesting paper in IJMS, demonstrating that the virus BoHV-1 could infect a human adenocarcinoma cell, leading to DNA damage and tumor growth. In this follow up study, they provide additional experiments regarding DNA damage caused by the same virus in the same cell line. They then go on to show that beta-catenin, which is known to have important roles in genomic stability following viral infection, is also important in this specific cell model of viral infection. While the experiments performed here seem to be of a good technical standard, I must unfortunately conclude that the novelty value of this paper is extremely low. These experiments could easily have been included in their 2021 paper, and do not justify an additional IJMS paper in their own right.

Technically, my major concern regarding the paper is the Western blotting, and in particular how it is quantified. The method by which the blots are visualized is not clearly stated (beyond the use of “antibodies”) and the ImageJ method they use for quantification is only of use if signals are not saturated – that is if the protein level is proportional to the signal.

In all quantified blots, it is crucial that non-saturated signal is shown and used for quantification. Most signals in figure 1A appear to be saturated, as are the beta-actin signals in 3C, 7C, 7D and possibly 1D and beta-catenin in 4B. Additionally, for all quantified Western blots, please supply a graph with error bars so that consistency over experiments can be seen (rather than just a single “fold change” number).

For all blots, supply molecular weight markers, and ideally use black borders to mark the edge of the blots.

Finally, in figure 4, the additional catenin bands seen are extremely weak. They may be specific, however I would like to see additional exposure times shown (possibly on a gel in which the bands have been separated more) to be sure, along with replicates to demonstrate that it is consistent over multiple experiments.

Author Response

Reviewer #1:

Less than 6 months ago, the laboratory of Liqian Zhu published an interesting paper in IJMS, demonstrating that the virus BoHV-1 could infect a human adenocarcinoma cell, leading to DNA damage and tumor growth. In this follow up study, they provide additional experiments regarding DNA damage caused by the same virus in the same cell line. They then go on to show that beta-catenin, which is known to have important roles in genomic stability following viral infection, is also important in this specific cell model of viral infection. While the experiments performed here seem to be of a good technical standard, I must unfortunately conclude that the novelty value of this paper is extremely low. These experiments could easily have been included in their 2021 paper, and do not justify an additional IJMS paper in their own right.

Response: We would like to thank the reviewer for assessing our manuscript and appreciate the comments and valuable suggestions. We have conducted additional modifications according to most of the suggestions.

Technically, my major concern regarding the paper is the Western blotting, and in particular how it is quantified. The method by which the blots are visualized is not clearly stated (beyond the use of “antibodies”) and the ImageJ method they use for quantification is only of use if signals are not saturated – that is if the protein level is proportional to the signal.

Response: The methods on how to develop and quantify the immunoblots have been added to the text in the revised manuscript. We understand the concerns that we cannot use overexposed immunoblots. Actually, we have most of the immunoblots with both overexposure and light exposure. The latter are used for intensity. We showed most of the blots with longer exposure because we thought those blot looks better. Now we have rearranged the figures and presented blots with light exposure.

In all quantified blots, it is crucial that non-saturated signal is shown and used for quantification. Most signals in figure 1A appear to be saturated, as are the beta-actin signals in 3C, 7C, 7D and possibly 1D and beta-catenin in 4B. Additionally, for all quantified Western blots, please supply a graph with error bars so that consistency over experiments can be seen (rather than just a single “fold change” number).

Response: Thanks for the suggestions. Now we have replaced most of the immunoblots appeared to be saturated with light exposure ones. Furthermore, graphs with errors derived from at least three independent experiments were shown for all the immunoblots that need quantitative analysis.

For all blots, supply molecular weight markers, and ideally use black borders to mark the edge of the blots.

Response: Thanks for the suggestions. We have modified all the blots according to the suggestions.

Finally, in figure 4, the additional catenin bands seen are extremely weak. They may be specific, however I would like to see additional exposure times shown (possibly on a gel in which the bands have been separated more) to be sure, along with replicates to demonstrate that it is consistent over multiple experiments.

Response: Actually, we are cautious about this faint band, and have repeated more than three times. Please see the following images.

Reviewer 2 Report

The manuscript is based on thorough research of the scientific literature and shows that BoHV-1 infection disrupts 53BP1-mediated DNA damage repair and that β-catenin is a potential host factor restricting both virus replication and DNA damage in A549 cells. The results presented prove that β-catenin is involved in the restriction of viral protein expression and virus-induced DNA damage in A549 cells. The conclusions are supported by the data presented, this paper being of interest considering the original approach on the very hot topic addressed.

Generally, the quality of the article is good and the manuscript is intriguing for the readers.

This study has an interesting approach but requires extended studies on more tumor cell lines and patients to confirm the β-catenin role in the restriction of viral protein expression and virus-induced DNA damage.

Overall, I consider the article could be a useful contribution to the journal. I recommend the manuscript for publishing after minor changes and updates have been taken into consideration by the authors. There are some typographical errors in the manuscript that I have highlighted.

My observations are highlighted in the manuscript:

  1. The introduction could be improved by explaining the role of β-catenin in virus-produced infection and DNA damage induced by a viral infection in other tumor cell lines than in A549 cells.
  2. The methods used require a more detailed description of the steps taken.
  3. Some of the sentences I have highlighted in the manuscript are difficult to understand and need to be rewritten.
  4. The explanations in the legends of Figures 1, 4, and 6 need to be improved so that they can be better understood.

Author Response

Reviewer #2

The manuscript is based on thorough research of the scientific literature and shows that BoHV-1 infection disrupts 53BP1-mediated DNA damage repair and that β-catenin is a potential host factor restricting both virus replication and DNA damage in A549 cells. The results presented prove that β-catenin is involved in the restriction of viral protein expression and virus-induced DNA damage in A549 cells. The conclusions are supported by the data presented, this paper being of interest considering the original approach on the very hot topic addressed.

Generally, the quality of the article is good and the manuscript is intriguing for the readers.

This study has an interesting approach but requires extended studies on more tumor cell lines and patients to confirm the β-catenin role in the restriction of viral protein expression and virus-induced DNA damage.

Overall, I consider the article could be a useful contribution to the journal. I recommend the manuscript for publishing after minor changes and updates have been taken into consideration by the authors. There are some typographical errors in the manuscript that I have highlighted.

Response: We would like to thank the reviewer for assessing our manuscript and providing comments and valuable suggestions. It is really a good idea to verify the roles of β-catenin in limiting virus replication with more tumor cells as an independent study in the future. We have revised typographical and grammar errors according to the suggestions.

My observations are highlighted in the manuscript:

Response: We have revised typographical and grammar errors following the instructions marked in the manuscript.

The introduction could be improved by explaining the role of β-catenin in virus-produced infection and DNA damage induced by a viral infection in other tumor cell lines than in A549 cells.

Response: According to the suggestions, some contents regarding to the roles of β-catenin in DNA damage in a context either with or without virus infection have been added into the introduction, as well as the discussion. We only find one virus, HCV that induce DNA damage via β-catenin which is different from that observed in the other tumor cells that it promotes DNA damage.

The methods used require a more detailed description of the steps taken.

Response: According to the suggestions, detailed description about the methods has been added in the revised manuscripts.

Some of the sentences I have highlighted in the manuscript are difficult to understand and need to be rewritten.

Response: According to the suggestions, these sentences have been modified.

The explanations in the legends of Figures 1, 4, and 6 need to be improved so that they can be better understood.

Response: According to the suggestions, they have been modified.

Round 2

Reviewer 1 Report

The authors have done a good job of addressing the technical issues which I raised. With the revised figures, it is clear that the experiments have been done to a high standard. I therefore have no scientific concerns with the manuscript.

I still believe, however, that the novelty value of the paper is extremely low. These experiments would all have fit well into the authors publication from 6 months ago, but I do not believe they warrant an additional paper on their own. For this reason, and this reason only, I have selected “Reconsider after major revision” and will leave it to the editors to make their decision.

Author Response

Reviewer #1

The authors have done a good job of addressing the technical issues which I raised. With the revised figures, it is clear that the experiments have been done to a high standard. I therefore have no scientific concerns with the manuscript.

I still believe, however, that the novelty value of the paper is extremely low. These experiments would all have fit well into the authors publication from 6 months ago, but I do not believe they warrant an additional paper on their own. For this reason, and this reason only, I have selected “Reconsider after major revision” and will leave it to the editors to make their decision.

Response: We would like to thank the Reviewer for taking precious time to assess our manuscript again and appreciate the comments and valuable suggestions. However, we cannot agree with the Reviewer on the low novelty value, and it could easily be included in our 2021 paper, published at 6 months ago. Our current study and the previous study published six months ago (Qiu et al., 2021, Int J Mol Sci) address two different questions. The previous study is to establish that BoHV-1 has the potential to be used as a therapeutic oncolytic virus using a xenograft mouse model. We have shown that BoHV-1 kills tumors by inducing DNA damage to support the role of BoHV-1 as a therapeutic oncolytic virus. However, we do NOT know the mechanisms of how BoHV-1 causes DNA damage and how host factors regulate this process. The current study is to answer these questions by pinpointing the molecular mechanisms. Here, for the first time, we provide evidence showing that BoHV-1 infection disrupts 53BP1-mediated DNA damage repair, and β-catenin is a potential host factor restricting both virus replication and DNA damaging in A549 cells. Thus, this study is a totally new story with novel mechanisms, which cannot and should not be added to our previous paper.

Based on the above reasons, we think that the findings in this manuscript are of high novelty and deserve to be published in Int J Mol Sci.

Round 3

Reviewer 1 Report

I have read the authors reply, but I am afraid that my opinion is unchanged. I accept that there is some novelty to this paper, but the progress is incremental relative to their last paper which was published only a short time ago. In light of this very recent paper, I just cannot see that the additional experiments here warrant another publication in an impact factor 6 journal like IJMS. Ultimately, however, this is an editorial decision.

That said, I believe that the experiments done here are technically good, and the general field of research is interesting. I would be happy to review again if the authors either expand the scope of this manuscript in the future, or resubmit it to a more appropriate journal.